# Wolf diet and prey selection in the South-Eastern Carpathian Mountains, Romania

**Teodora Sin** [1,2]*, **Andrea Gazzola**[2], **Silviu Chiriac**[3], **Geta Rîşnoveanu**[1]*

**1** Department of Systems Ecology and Sustainability, Faculty of Biology, University of Bucharest, Bucharest, Romania, **2** Association for the Conservation of Biological Diversity, Focşani, Vrancea County, Romania, **3** Environmental Protection Agency, Focşani, Vrancea County, Romania

* teodora.sin@g.unibuc.ro (TS); geta.risnoveanu@g.unibuc.ro (GR)

**Data Availability Statement:** All relevant data are within the paper and its Supporting Information files.

**Funding:** Putna-Vrancea Natural Park Administration (http://parculputnavrancea.ro/),

## Abstract

The Romanian wolf population, one of the largest in Europe, occupies a total home-range of 154500 km$^2$ and is spread across a variety of landscapes–from anthropized hills and plateaus to remote, densely forested mountains. However, this population is markedly understudied, and even basic knowledge of the species' feeding habits is deficient. Wolf diet was assessed based on 236 scat samples collected between November 2013 and October 2014, by following pre-established transects (total length = 774 km). The study area (600 km$^2$) is a multi-prey ecosystem in the southern sector of the Eastern Romanian Carpathians. Our results emphasize that more than 80% of the wolf diet is based on wild ungulates. The wild boar is clearly selected (D = 0.74) and is the most common species in the diet (Bio = 72%), while roe deer (Bio = 10%) and red deer (Bio = 5%) have a smaller contribution. Domestic species represented the second-largest prey category in both seasons. Among them, dog is a particularly important source of food (Bio 3.5–10.9%). Other domestic species (goat, sheep, horse) have marginal importance in the wolf diet and seasonal occurrence. Standardized niche breadths are low in both seasons ($B_{Aw}$ = 0.07, $B_{As}$ = 0.12), and a high degree of overlap in the resources used has been observed ($Ô_{ws}$ = 0.99). Our study represents the first step towards understanding the wolf foraging behaviour in the Romanian Carpathians and is valuable to address the complex issues of wolf and wild ungulate population management and conservation.

## Introduction

The wolf (*Canis lupus*, L. 1758) is commonly described as a generalist species, as a result of its historically wide distribution and its diverse and adaptable feeding behaviour [1]. In Europe, recent reviews revealed that wolf feeds mainly on medium-sized wild ungulates, such as wild boar *Sus scrofa*, roe deer *Capreolus capreolus* and chamois *Rupicapra rupicapra* or large-sized wild ungulates, such as reindeer *Rangifer tarandus*, elk *Alces alces* and red deer *Cervus elaphus* [2, 3]. Although evolutionary the wolf is well-adapted to catching large herbivores, when they are not available, it can consume anything from small-sized wild mammals to fruits, birds and anthropogenic resources, such as garbage and livestock [4–6]. Because of this feeding behaviour, the wolf has always been perceived by humans as a competitor (for game species or

Association for the Conservation of Biological Diversity (www.acdb.ro) and the Environmental Protection Agency, Vrancea County (http://apmvn.anpm.ro/) provided logistic support during data collection. Part of staff costs for TS and laboratory work has been supported from the strategic grant POSDRU/159/1.5/S/133391, Project "Doctoral and Post-doctoral programs of excellence for highly qualified human resources training for research in the field of Life sciences, Environment and Earth Science" co-financed by the European Social Fund within the Sectorial Operational Program Human Resources Development 2007-2013 (http://fondurieuropene.centre.ubbcluj.ro/posdru1591-5g133391-programe-doctorale-si-post-doctorale-de-excelenta-pentru-formarea-de-resurse-umane-inalt-calificate-pentru-cercetare-in-domeniile-stiintelor-vietii-mediului-si-pamantului/). Additional support for TS has been provided by Pro Biodiversitas SRL. The funders had no role in study design, data collection and analysis, decision to publish, or preparation of the manuscript.

**Competing interests:** The authors have declared that no competing interests exist.

livestock) and has been persecuted under a continuously reinforced conflict, manifested at different intensity across the wolf distribution range [7].

Nowadays, at the European level, habitat loss and fragmentation have led to a lack of suitable habitat and a patchy distribution, with several wolf populations of various sizes and degrees of isolation [8]. The more fragmented and human-dominated landscapes of Central-Western Europe provide habitats for small, isolated populations, while some of the largest wolf population can be found in Eastern Europe [9], especially in the Carpathian mountains where forested landscapes and biodiversity have been historically better preserved [10, 11]. However, since the fall of the communist regime in the late 1980s, the ecological integrity of these forests decreased continuously under the pressure of inefficiently controlled grazing and increasing tourism activities as well as recent changes in land ownership and forestry practices [11, 12].

Habitat particularities, prey density and vulnerability [13], as well as the level of natural resources exploitation by human [14, 15] are considered major key factors that influence the spatial and temporal dynamics and feeding ecology of the wolf. Zlatanova et al. [2] noted that wolves show specific ecological adaptations in their diet, depending on whether they inhabit natural or anthropogenic habitats. Concurrently, the availability of prey and wolf sensitivity to prey changes determine the ecological status of a wolf pack [15]. High variability of prey selection might occur locally (i.e. between neighbouring packs), and regionally [16], in relation to the particular behavioural and ecological context [1].

Based on the data provided by Kaczensky et al. [9], it was assessed that the wolf population on the Romanian territory represents more than 80% of the entire Carpathian wolf population. Actively managed in the past, the wolf disappeared from the Romanian lowlands by the end of the 1970s [17], and it is now found only in the Carpathian Mountains and the Transylvanian Plateau [18], where it occupies an area of 154500 $Km^2$. Across this area, the species is found in a variety of landscapes, among which the remote, densely forested mountains represent more than 70%, while highly anthropic hills and plateaus form less than one-third of the wolf territory.

Wolf diet and wolf-prey interactions have been extensively studied in Europe. However, few studies have documented the feeding ecology of the Romanian wolf population, which is the most representative part of the Carpathian wolf population. Although some studies exist (e.g. [19, 20], the information is sporadic and inexhaustive. Based on stomach content analyses, in the 1970s wild and domestic ungulates occurred in similar proportions (28% and 27%, respectively) [19], while more recently higher percentages of wild ungulates (53%) were reported [20].

The current study represents the first contribution to wolf diet in Romania based on a standardized data collection and analysis of non-invasively collected wolf scats. It was conducted in a wild and compact remote forested area where four species of wild ungulates (red deer, roe deer, wild boar and chamois) occur at low densities and livestock is abundant during summer.

The aims of our study are to: i) assess the contribution of different food items in the wolf diet; ii) evaluate the seasonal variability in wolf feeding habits. Given the low densities of wild ungulates in the area, we hypothesized that wolves would display opportunistic behaviour, feeding on a broad spectrum of food items. In winter, we would expect wild ungulates to be consumed proportional to their abundance in the habitat. On the contrary, in summer the wolf diet would shift mainly to domestic ungulates as they become more accessible.

## Materials and methods

### Ethics statement

Our research did not involve capture, handling or killing of animals, therefore did not require the approval of animal care and use procedures. Permissions for field studies were given by the Putna-Vrancea Natural Park Administration and Local Environmental Protection Agency.

## Study area

The study was conducted in the Southern sector of the Eastern Romanian Carpathians (N 45.915, E 26.502) (Fig 1), over a surface of 600 km$^2$, protected under the legislative framework of the Natura 2000 network (ROSCI0208, ROSCI0395, ROSCI0130). Elevations range from 491 m to 1785 m a.s.l., and the terrain is characterised by narrow valleys and steep, rugged slopes. Snow cover lasts from 62 to 170 days a year, depending on the altitude, and the average snow depth is 51 cm (in January–February). Compact forest habitats (53% mixed, composed by *Picea abies*, *Fagus sylvatica* and *Abies alba* or *F. sylvatica* and *A. alba;* 28% coniferous, mostly *P. abies*; 11% broad-leaved *F. sylvatica* and *Carpinus betulus* or *Fraxinus excelsior*, *Acer pseudoplatanus* and *Ulmus glabra*) dominate the landscape, covering more than 92% of the study area. Pastures and natural grasslands cover only 7% of the area, while artificial and agricultural land covers less than 1% (assessment based on CORINE Land Cover 2012 [21]).

Besides the wolf, several carnivore species like the European lynx (*Lynx lynx*), brown bear (*Ursus arctos*), fox (*Vulpes vulpes*), wildcat (*Felis silvestris*), European badger (*Meles meles*),

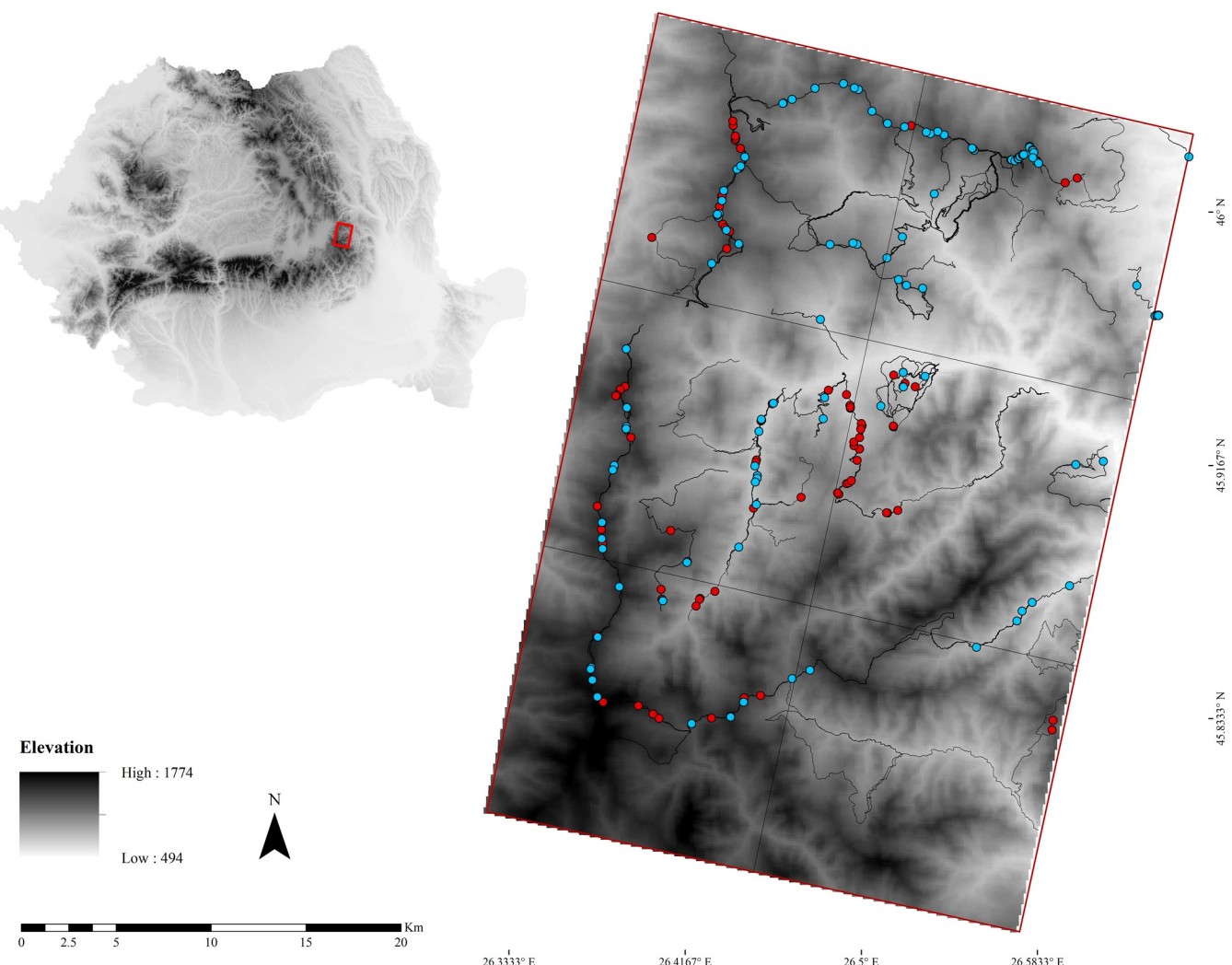

**Fig 1. Study area.** Location of the study area in the South-Eastern Romanian Carpathians; sampling units (grid), the location of transects (lines) and collected samples in summer (red dots) and winter (blue dots).

pine marten (*Martes martes*) and stone marten (*Martes foina*) are present in the region. A complex ecological community comprising a wide range of prey species known to occur in the diet of wolves, such as red deer (*Cervus elaphus*), roe deer (*Capreolus capreolus*), wild boar (*Sus scrofa*), chamois (*Rupicapra rupicapra*) and small rodents characterise the area.

Permanent human disturbance is limited to only two small-sized linear settlements located in the north-central part of the study area. Primary road density is less than 0.1 km/km$^2$. Hunting is allowed for both ungulate and large carnivore species and is based on annually established quotas. The logging activity takes place all year round, and over time it has led to a significant increase of the density of forest roads (5.2 km/km$^2$) [22]. According to transhumant tradition, grazing is practised from May to September, in 18 herding units with different livestock species (sheep, goats, cattle, and swine), horses and donkeys (unpublished data, LIFE13-NAT/RO/000205). The herds are placed on higher pastures, and over the night shepherds keep them outside, in fenced enclosures, usually defended by 1–2 guardian dogs and several small-sized mixed-breed dogs.

## Sample collection

A total of 236 scat samples were analysed (115 in winter, 121 in summer). As no reliable estimates of wolf numbers or pack distributions were available for the study area, a spatially balanced sampling strategy was used to increase the probability of detecting and collecting a representative number of samples. Considering the large home ranges and the daily long-distance movements of large carnivores, as well as the EU recommendations towards the facilitation of a standardized data collection procedure across Europe [23], six sampling units were defined by placing the 10x10 km EEA grid over a topographical map of the area. A set of transects randomly selected along the existent network of forest roads and footpaths was surveyed, by foot, across each of the six sampling units. The total transect length amounted to 774 km, and mean transect length was 8.8 ± 4.5 km. From November 2013 to October 2014, sampling units were visited between two to six times per season, the most visited sites being those where higher wolf presence was observed (see S1 Table).

To account for possible changes in wolf hunting and feeding behaviour our diet investigation was conducted during two biological seasons: November to April, hereafter referred to as *winter*, and May to October, hereafter referred to as *summer*. To correctly assign samples to each season, visual estimates of the deposition time based on the shape and structure of the sample, and environmental conditions such as snow cover, rain or sun exposure were considered. Samples were classified as "fresh", if deposition time was within a week, "medium", if deposition time was between seven days to a month, and "old", if scats seemed to be older than a month, generally those that were found under old snow layers. To avoid possible bias in diet estimates due to the collection of wolf scats at different sites (i.e. along roads, wolf travel routes, kill sites, denning areas or rendezvous sites) [24] only scats collected along the transects were considered in the analysis. A multi-criteria approach [25, 26] was used to reduce the chances of including scats of nontarget species in the diet estimate. Wolf and fox scats were distinguished based on their morphometry (shape, size). For wolf and dog scats, which are usually hard to differentiate when both species co-occur [25], additional control measures were applied. During the wintertime, scat content and odour were enough to assign the scat to a species because dogs are present only in villages or, when they sporadically occur throughout the forest, they are usually accompanying humans (loggers, hunters). In summer, scats collection was kept outside of an arbitrarily selected two km radius around active sheepfolds, to avoid collecting scats of shepherd dogs roaming in the nearby forests. Sheepfold areas inside the buffer zone were visited in May, before the arrival of the shepherds, and revisited in

October, after their departure, when only fresh scats were collected. If doubt about identification persisted, the scats were not included in analyses. The DNA analyses of 187 scat samples collected by the same observers in a subsequent study confirmed their ability to discriminate between wolf and other carnivores scats, with 97% (n = 181) of the samples being correctly attributed to wolf. All samples were preserved in sealed plastic bags, at constant temperature (4˚C), away from direct sunlight.

## Sample preparation and laboratory analysis

Sample preparation followed the standard procedures thoroughly discussed in previous papers [27, 28]. To break down the scats and remove all debris samples were soaked in water with detergent, then washed under running water while being filtered through sieves with 0.5 mm meshes. The remaining materials were air-dried for 24–48 hours. Once dried, undigested remains of prey (hairs, hoofs, claws, bone fragments) and plant materials (seeds, leaves, others) were placed in labelled plastic bags, along with a corresponding standard form, and stored until analysis. Prey items were identified by macroscopically examining hair remains and comparing them against a reference collection of mammal hairs. Blind tests were applied on randomly selected samples from the available collection of hairs of wild and domestic mammal species present in our study area to assess the ability of the two observers to identify the prey species. The process was repeated at least three times, and a species was considered to be accurately determined if the responses of both observers matched in 11 out of 12 cases (92%). To further increase the reliability, all the collected samples were cross-checked by both observers. When necessary, microscopic observation of hair structure (medulla and cuticula) was used [29–31]. If a species was not identified by any of the methods, it was recorded as "undetermined".

## Data analysis

**The contribution of different food items in the diet.** The percentage frequency of occurrence of different items in the diet (%Occ) was calculated based on the equation: %Occ = $N_i \div N_t \times 100$ where $N_i$ is the number of occurrences of food item"$i$", $N_t$ is the total number of occurrences of all food items [32].

As %Occ overestimates the importance of small preys [13], the contribution of each item to the total volume of scats (Vm) was assessed based on the equation: %Vm$_i$ = $\Sigma V_i \div \Sigma V_t \times 100$ where $V_i$ is the volume of prey "$i$" in each scat, and $V_t$ is the volume of all prey items in each scat [33]. The volume of specific prey items ($V_i$) in each scat was assessed by visually assigning them, using a reference grid, to one of the following fixed categories [28]: 0% (0–5), 25% (6–25), 50% (26–50), 75% (51–75), 100% (76–100). Certain food remains (hairs and leaves) with volumes less than 5% were discarded from the analysis (i.e. recorded as 0%), as they could have represented trace elements, accidentally ingested, or residuals from previous feedings [25, 28, 34].

Considering that the degree of digestion and digestibility is different for every food item [25], and that Vm overestimates the importance of prey when the percentage of scats containing only one item is high, Weaver's correction factor (Y) [35] was used to calculate the ingested biomass (Bio) and to estimate the actual contribution of each food item in the diet (hereinafter shown as %Bio), according to the following equations: $Y = 0.439 + 0.008 \times X$ where $Y$ represents the fresh mass (kg) of prey per scat, and X is the average mass of live prey, Bio = $\Sigma V_i \times Y$ and %Bio = Bio$_i \div$ Bio$_t \times 100$ where Bio$_i$ represents the ingested biomass of species $i$, and Bio$_t$ represents the ingested biomass of all species. Although assessing carnivore diet using biomass deals with the drawbacks of ranking food items by frequency and volume measures [25],

estimating ingested biomass of prey from scats is still prone to bias because the relationship between the fresh mass of prey per scat and prey body mass is influenced by some variables that in nature are hard to detect and measure. Among these we highlight: the age class or sex of prey, the amount and carcass body parts consumed by wolf, and the number of conspecifics feeding on the same carcass [36, 37]. Accurately discriminating between adults and young animals was not possible, therefore we used the average body mass of adult live prey species, as obtained from the literature (Table 1). To limit uncertainties affecting the method to calculate the biomass consumed from scats, sampling near kill sites was avoided. The consistency of the estimates between methods was evaluated using Kendall's coefficient of concordance ($W$).

To account for the effects of random sampling errors [27, 42], 95% bootstrap confidence intervals for mean %Bio (based on 2000 bootstrap replicates) were calculated for each prey item in the diet.

**Seasonal variability in feeding habits.**   Dietary diversity was assessed based on the standardized Levins' formula for measuring the niche breadth (B) [43]: $B = 1/\sum p_i^2$ where $p_i$ represents the proportion of the biomass of food item "$i$", as found in the estimated wolf diet, and $\sum p_i = 1$.

The niche breadth can take any value from 1 to n, "n" being the total number of food items found in the diet. A value of, or close to 1 represents a narrow niche breadth (or a high degree of specialization), while a value close, or equal to the total number of food items represents a broad niche breadth (or that the species is a generalist). The result was standardized on a scale from 0 to 1 using Hurlbert's formula [44]: $B_A = (B-1)/(n-1)$ where $B_A$ is the Levins' standardized niche breadth, B is Levins' measure of niche breadth, and "n" is the number of items in the diet.

Pianka's index of niche overlap ($\hat{O}_{ws}$) was used to quantify between season similarities in the diet: $\hat{O}_{ws} = \sum p_{iw} \times p_{is} / \sqrt{\sum p_{iw}^2 \times \sum p_{is}^2}$ where $p_{iw}$ and $p_{is}$ represent the proportion of the biomass of food item "$i$", as found in the estimated wolf diet in winter and summer respectively [45]. The index ranges from 0 (no overlap) to 1 (complete overlap).

The importance of different prey items in each season was compared using Kendall's coefficient of concordance (W), and between season item-specific differences were tested using randomization tests (5000 iterations). The last two analyses are based on biomass estimates.

**Table 1. Average body mass and total number of individuals of live prey items in the study area.**

| Species | Body mass (kg) | Reference | No. of individuals* |
|---|---|---|---|
| Wild boar | 66 | [38] | 962 |
| Red deer | 115 | [39] | 632 |
| Roe deer | 24 | [39] | 667 |
| Chamois | NA | NA | 145 |
| Sheep | 40 | [5] | 6500 |
| Goat | 30 | [5] | 230 |
| Horse | 234 | [25] | NA |
| Mustelids | 0.7 | [40] | NA |
| Small rodents | 0.06 | [25] | NA |
| Fox | 5.4 | [41] | NA |
| Dog | 22 | [25] | 173 |

*the numbers of wild ungulates were obtained based on Pellet Group Count surveys (performed in May) and those of domestic species were based on shepherd questionnaires. Reported numbers of domestic items are representative only for summer. No data are available (NA) for the rest of the species. See text for more details.

To verify if wolves exhibited a particular preference or avoidance of any of the wild ungulate species present in the habitat, Ivlev's electivity index, modified by Jacobs [46], was calculated based on the equation: $D = (p_i—a_i) / (p_i + a_i - 2p_ia_i)$ where $p_i$ and $a_i$ represent the proportion of the biomass of food item "$i$" as found in the estimated wolf diet and in the habitat respectively. The values of the index range from -1 to 1, with negative values indicating prey avoidance or inaccessibility, zero showing that prey are randomly consumed, and positive values indicating wolves are actively selecting a specific prey. Prey selectivity was assessed only in the winter season, as our estimates of relative abundance of prey species were obtained in spring before the calving season occurred.

Values in Table 1 were used to estimate the biomass of ungulates available in the habitat. The data represents the most recent estimates of ungulate abundance available for our study area. For wild ungulates, a systematic survey using the Faecal Standing Crops approach (Faecal Pellet Group Counts) was applied. In May 2015, Faecal Standing Crops were measured in sixty strip transects (10 strip transects per sampling unit) with a 2 m fixed-width and 150 m fixed-length. The following conditions were considered for the distribution of the strip transects: i) starting points located 50 m away from forest roads; ii) one-kilometre distance between transects; iii) randomly determined direction. The population size of domestic prey was assessed based on questionnaires applied to shepherds from May to September 2015. Population estimates were converted to biomass by multiplying prey numbers with the average body mass of adult live prey items (Table 1). All data analyses were performed with R software [47].

## Results

### The contribution of different items in the diet

Overall, the wolf diet comprised 11 different food items (Table 2). More than 95% of the samples contained one item, and the maximum number of items per sample was two. Wild ungulates represented over 80% of the diet (%Occ = 82.59, %VM = 83.16, %Bio = 87.62), and among the three species of wild ungulates found in the samples, wild boar was the dominant prey, with roe deer being the second most important, and red deer the last (Table 2). Domestic species were identified in 34 samples, representing 13.77% in terms of occurrence, 14.09% in volume, and 11.22% in biomass. Dogs had the highest occurrence of all identified domestic species, representing 7.12% of the total biomass consumed by wolves. The overall occurrence of livestock was low (%Occ = 4.1), and only three species (goat, sheep, horse) were identified in the collected samples. Small- and medium-sized mammals had a marginal occurrence in the diet (%Occ = 2.43), accounting for about 1% of the biomass consumed by wolves (Table 2). In terms of prey ranking, diet estimates were consistent across all methods ($W = 0.95$, $X^2 = 25.9$, p = 0.002).

### Seasonal variability in feeding habits

Standardized niche breadths were 0.07 in the winter, and 0.12 in the summer. Between seasons, resource use overlap was 0.99 and the ranks of prey items did not differ significantly ($W = 0.83$, $X^2 = 15.08$, $p = 0.08$).

Wild ungulates remained the primary prey category in both seasons (%Bio$_w$ = 93.9, %Bio$_s$ = 81). Although significantly lower in summer ($p_{randomization} = 0.04$), wild boar was the main prey species in both seasons (Fig 2). The relative biomass of red deer in wolf diet decreased in summer, while the roe deer remained constant (Fig 2). The consumption of domestic species increased during the summer (%Bio$_w$ = 5.66, %Bio$_s$ = 17.07), significant differences being observed for goat ($p_{randomization} = 0.03$) and dog ($p_{randomization} = 0.02$). Besides the goat, other three new food items have been observed in the summer diet (sheep %Bio$_s$ = 1.72, fox %Bio$_s$ =

**Table 2. Annual wolf diet in the South-Eastern Romanian Carpathians, from November 2013 to October 2014 (n = 236).**

| Prey category/Food item | Scat no. | %Occ | %Vm | %Bio | 95% CI of %Bio | |
|---|---|---|---|---|---|---|
| | | | | | lower limit | upper limit |
| **Wild ungulates total** | **204** | **82.59** | **83.16** | **87.62** | **77.31** | **98.55** |
| 1. wild boar | 159 | 64.37 | 65.57 | 72.19 | 65.3 | 78.38 |
| 2. roe deer | 36 | 14.9 | 14.21 | 10.2 | 7.22 | 13.62 |
| 3. red deer | 8 | 3.32 | 3.38 | 5.23 | 1.96 | 9.79 |
| cervids undetermined* | 1 | / | / | / | / | / |
| **Domestic species total** | **34** | **13.77** | **14.09** | **11.22** | **8.02** | **16.33** |
| 4. dog | 24 | 9.72 | 10.17 | 7.12 | 4.46 | 9.79 |
| 5. goat | 5 | 2.31 | 2.42 | 1.87 | 0.66 | 3.72 |
| 6. sheep | 2 | 0.93 | 0.97 | 0.84 | 0.09 | 2.31 |
| 7. horse | 2 | 0.81 | 0.53 | 1.39 | 0 | 6.14 |
| domestic undetermined* | 1 | / | / | / | / | / |
| **Small- and meso- mammals total** | **6** | **2.43** | **2.23** | **1.16** | **0.42** | **2.62** |
| 8. fox | 2 | 0.81 | 0.85 | 0.47 | 0 | 1.16 |
| 9. mustela sp. | 2 | 0.81 | 0.85 | 0.43 | 0 | 1.06 |
| 10. small rodents | 2 | 0.81 | 0.53 | 0.26 | 0 | 1.11 |
| **Other** | | | | | | |
| 11. plant material | 3 | 1.21 | 0.53 | / | / | / |

%Occ = the relative frequency of occurrence, %Vm and %Bio represent the relative volume and biomass of prey categories identified in the scat analyses; 95% confidence intervals (CI) of the relative biomass of different prey categories are shown.

*cervids and domestic undetermined were split proportional to their appearance in the diet between the known species of the same family.

0.95, and small rodents %$\text{Bio}_s$ = 0.54). *Martes* sp. had a marginal occurrence in both seasons (%$\text{Bio}_w$ = 0.42, %$\text{Bio}_s$ = 0.44).

When compared to the available prey biomass, wild boar occurred in the wolf diet more frequently (D = 0.74), while red deer less frequently (D = -0.85) than available in the habitat. Roe deer was used proportional to its availability (D = 0.01) (Fig 3). No chamois was found in the analysed samples

## Discussion

### The contribution of different items in the diet

Our results emphasize that more than 80% of the wolf diet is based on wild ungulates, and contrary to our hypothesis, the niche breadth is narrow. As opposed to the other two studies undertaken in Romania [19, 20], we have found a diminished importance of domestic ungulates in the wolf diet. The domestic to wild prey shift follows the tendency of wolf diet changes across Europe [48] and is generally attributed to an increase of wild prey abundance and a decline of human activities in mountain areas [49]. In Romania, these factors may only partly explain the changes. The abundance of both wild ungulates and wolf increased over the past four decades. Nevertheless, based on the official raw data reported by responsible Romanian authorities, the prey to wolf ratio decreased in all cases except for wild boar (see S2 Table). Moreover, human pressure has modified wolf habitat only to a small extent. The forest surface remained the same, but the trees age classes and species diversity reduced as a response to past and present management practices [50]. The abandonment of permanent settlements and croplands in the mountain region [51] considerably reduced the presence of people in the forests and possibly modified human-wildlife overlap patterns and temporal use of space and

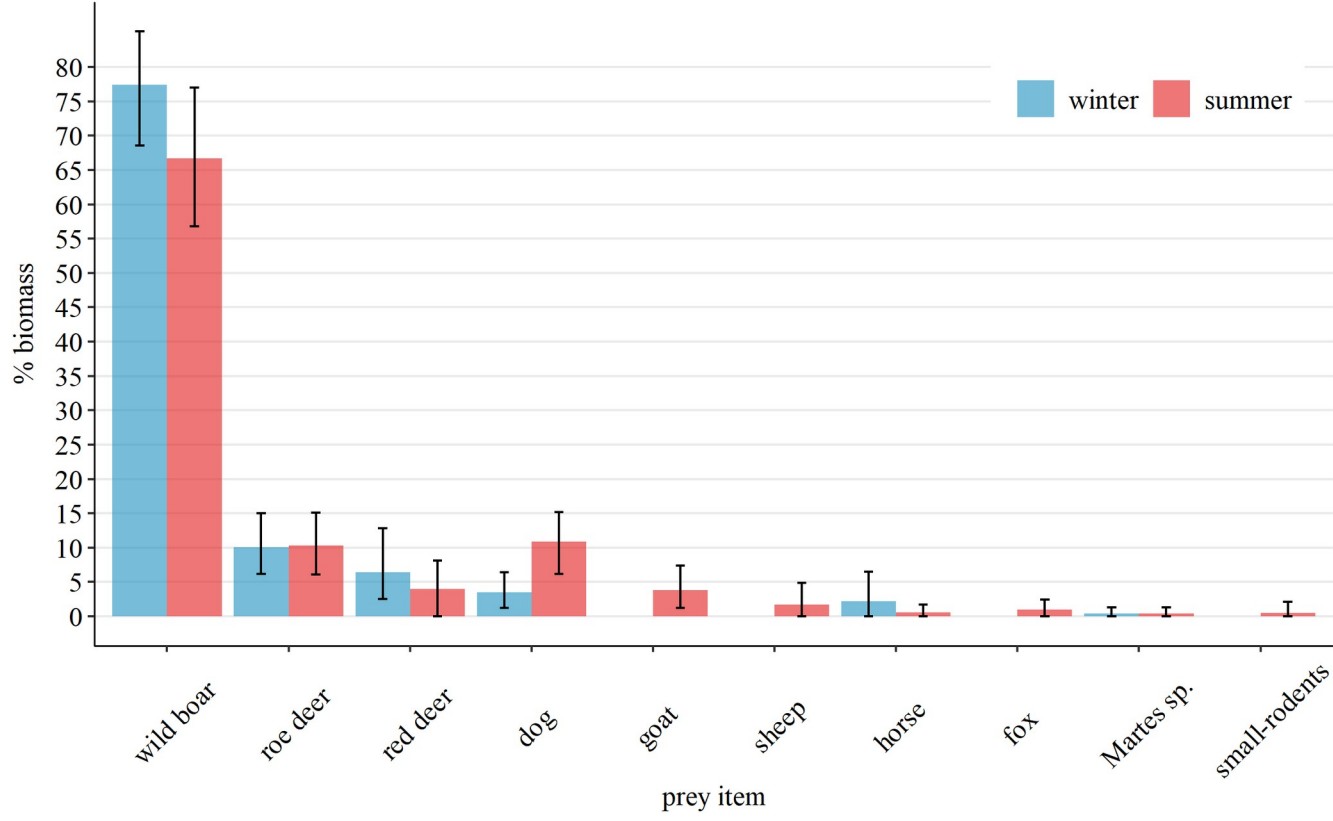

**Fig 2. Seasonal change in the biomass consumed by wolves in the South-Eastern Romanian Carpathians.** Winter: November 2013-April 2014, Summer: May-October 2014. Error bars indicate 95% confidence intervals.

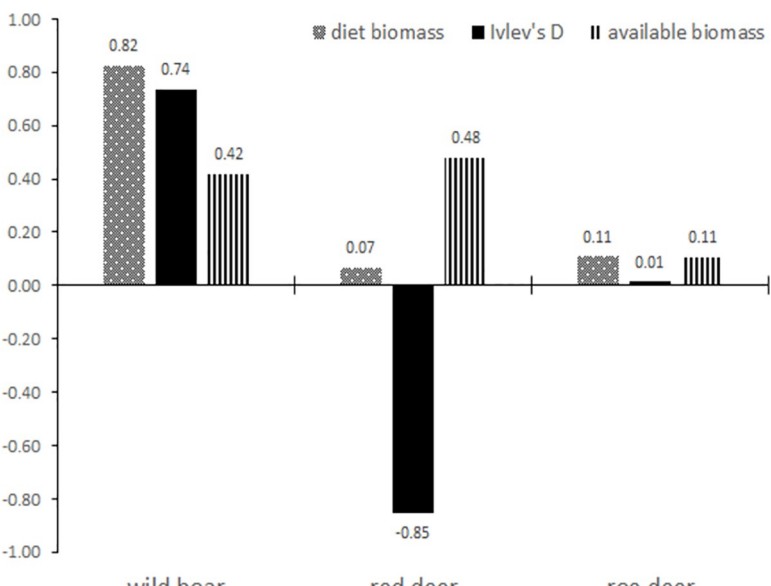

**Fig 3. Prey selectivity by wolves in the South-Eastern Romanian Carpathians.** D = Ivlev's index of selectivity which has values between 0 and 1; diet biomass–the relative biomass as present in the diet; available biomass–as present in the wild habitats.

resources by prey and predators. The depopulation process has led to a drastic decline of shepherding practices [52], also observed in the livestock to wolf ratio, which at the national level has reduced by more than a half (see S2 Table).

Species-wise, we found the wolf diet in the South-Eastern Carpathian Mountains to be similar to that of Southern Europe [2], also characterized by high consumption of wild boar [53]. On the contrary, in most of the Central-Eastern European countries cervids (red deer and roe deer) are the most common species in the diet [2]. In our study area, densities of the three wild ungulate species are similar, with wild boar being only slightly more abundant than the other (Table 1). Still, the latter is clearly selected, whereas red deer is avoided and roe deer is consumed proportional to its availability in the environment.

Species rankings were preserved across all analysis methods, supporting the important contribution of wild large-prey in the diet, and the occasional consumption of small mammals. An optimal foraging strategy would imply that wolves prey upon the species that ensure the most energy intake and the least energy expenditure [54]. The levels of energy spent to handle prey are low when prey is abundant (therefore encounter rates are high) or vulnerable, and the probability of a successful outcome upon attack is significant. Given the densities of wild prey in our region, their abundance alone cannot justify why the wild boar is so highly used [53]. Since the energy intake does not differ among ungulates [55], prey vulnerability may have a more considerable influence on the wolf prey choice.

Predictability of distribution and group size are generally associated with increased prey vulnerability [48, 56]. The wild boar is more gregarious than deer, it uses dense vegetation areas as resting sites, and moves on relatively fixed trails between feeding and resting sites [57, 58]. Unlike the red deer and roe deer, it has larger litters and the adult to juvenile ratio is in favour of the latter, with piglets and yearling wild boars being the most vulnerable to predation [39, 59]. Climatic factors may also play a role in shaping the feeding habits of the wolf. The snow layer in the South-Eastern Carpathian Mountains is thick (50 cm on average) and compact, due to a sizeable day-night temperature gradient. Deep snow, of over 70 cm, has been shown to affect moose escape speed and increase wolf hunting success [60]. It may also affect wild boar escape speed, in contrast to the red deer, which is supposedly able to flee faster given the high stature and length of the limbs [59]. Concurrently, temperatures below zero (-20°C) determine the freezing of the soil, which results in an inability of wild boar to find food. Undernutrition was one of the main wild boar mortality factors in Białowieża National Park [59], suggesting that the high occurrence of wild boar in the diet may not necessarily be due to active predation.

Besides abundance and the degree of aggregation, the use of space may also influence encounter rates and trophic interactions. Hunting of ungulates by humans changes prey behaviour and habitat use, making them search for better cover even in habitats where natural predators do not exist [61, 62]. In our study area, hunting occurs during the legal periods for roe deer and red deer, while hunting of wild boar males is allowed all year round. Concurrently, logging occurs throughout the year, and the dense network of forest roads facilitates human access all across our study area. The continuity and high intensity of human movement along trails may contribute to changes in prey and wolf behaviour and even limit available habitat [63]. Currently, the human-prey-predator interaction in our area is unassessed, and it should form the object of thorough studies.

Chamois, although present in our study area, did not occur in the analysed samples. The densities are very low, and its defence reaction is to escape into steep terrain, inaccessible to the wolf. Domestic ungulates only appear as accessory species, with an increased use during summer.

## Seasonal diet

The narrow seasonal niche breadths and the high diet overlap between the two seasons emphasize the high degree of specialization of the wolf in our study area. Altogether, the three species of wild ungulates found in the wolf diet in our region made up 94% and 81% of the total food biomass in winter and in summer respectively.

Contrary to the expected seasonal diet shift from wild to domestic ungulates, we did not observe a noticeable change in feeding habits, and wild boar still represented the main prey consumed by the wolf in both seasons (Fig 2). In other parts of the Carpathians, high use of wild boar has been observed in winter, and it has been reported to drop considerably during summer [64, 65].

In both seasons, domestic species represented the second-largest prey category, with dogs being the most important food item. Although the consumption of dogs by wolves is not a new discovery, its prevalence in the wolf diet in our study area is high when compared with other areas in Europe [25, 66, 67]. In human-dominated landscapes, wolf and dog ranges overlap considerably and dogs can sometimes be a profitable source of food for the wolf [68]. In our study area, dogs are less accessible in winter, because they are found in or near settlements. In summer, their numbers increase following the arrival of shepherds in the high pastures. Guardian shepherd dogs are rarely used to guard the sheep, and small-sized mixed-breed dogs are prevalent. Besides availability, their size and predictable location (fixed sheepfold locations) makes them a vulnerable and easily accessible species.

## Management and conservation implications

This study is important because it provides useful information to inform wildlife management decisions. Many of the management actions planned so far have been carried out in the absence of previous investigations.

According to the Romanian legislation, wolf culling is only allowed if the species produces damage to livestock. Considering our results, the low use of domestic ungulates in the wolf diet does not justify wolf culling, especially not at the national level, as it was the case before the large carnivore hunting ban stated in 2016 by the Ministry of Environment, Water, and Forests. The low use of domestic ungulates is also supported by the low number of successful attacks and reduced number of animals lost (<0.1% heads/sheepfold/grazing season) reported by shepherds in our study area (unpublished data, LIFE13NAT/RO/000205). While the situation may vary regionally, better damage prevention and functional compensation system would benefit the wolf population more and better serve the purpose of ensuring a favourable conservation status of the population.

On the other hand, by killing prey, hunters still commonly believe that wolf produces damage to economically valuable wild ungulates populations. As such, they consider controlling the wolf population a necessity and request much higher culling quotas than those assigned by the state authorities. Using culling as a management measure for the wolf has been shown to increase poaching as well [69, 70]. In Romania, Popescu et al. [71] suggest that these factors may have a much higher impact on the wolf populations than they initially assumed in their study. It is therefore necessary to study the impact of wolves on prey to quantify how much wolf consumes. This is fundamental to raise stakeholders' awareness and make informed governmental decisions. As the assigned quotas for wolf and prey species are currently not based on robust data, the unaccounted cumulative impact may have consequences on both prey and wolf populations.

Our findings regarding the high use of dog, especially during summer, suggest that dogs are a particularly accessible food source for wolves. Moreover, the widespread presence of dogs in

our study area can represent a threat to wolf populations by carrying disease and favouring the process of dog-wolf hybridization.

In Europe, prevention measures have been recognized as the most effective management tools to maintain an acceptable low level of livestock depredation and, consequently, to guarantee the wolf conservation in the long term [72].

In our study area, even if depredation is generally expected to increase during summer, based on the increased availability of livestock, the traditional protective measures used by shepherds appear to be effective against wolf attacks. Shepherds are permanently present near flocks, they use dogs and at night they keep flocks in enclosures, thus making them inaccessible to the wolf. Shepherding practices have remained the same for centuries in Romania as the wolf never disappeared from the Carpathians. This continuity of traditional livestock protection measures may have a significant contribution to the reduced number of successful depredation cases.

## Supporting information

**S1 Table. Detailed information on the entire data set used in the analysis: location of scat collected, sampling date, the month of scat deposition and the volume of food items in each sample.**
(XLSX)

**S2 Table. The prey to wolf ratio in 1970 and 2012, calculated based on species abundances estimated at the national level by the responsible Romanian authorities.**
(PDF)

## Acknowledgments

The help of WolfLife project (LIFE13NAT/RO/000205) volunteers and colleagues Andrea Corradini, Gabriela Rizzardini, Kieran O'Mahony, Guido Rastrelli during the fieldwork activities is very much appreciated. We thank Hannah Kirkland for the help with language editing and to the reviewers, Sandro Lovari and Maria Petridou, for the valuable suggestions and constructive comments made to improve this MS.

## Author Contributions

**Conceptualization:** Teodora Sin, Andrea Gazzola, Silviu Chiriac, Geta Rîşnoveanu.

**Data curation:** Teodora Sin.

**Formal analysis:** Teodora Sin, Andrea Gazzola, Geta Rîşnoveanu.

**Funding acquisition:** Silviu Chiriac, Geta Rîşnoveanu.

**Investigation:** Teodora Sin, Andrea Gazzola.

**Methodology:** Teodora Sin, Andrea Gazzola.

**Project administration:** Andrea Gazzola, Geta Rîşnoveanu.

**Resources:** Silviu Chiriac, Geta Rîşnoveanu.

**Supervision:** Andrea Gazzola, Geta Rîşnoveanu.

**Validation:** Teodora Sin, Andrea Gazzola, Geta Rîşnoveanu.

**Visualization:** Teodora Sin.

**Writing – original draft:** Teodora Sin.

**Writing – review & editing:** Andrea Gazzola, Silviu Chiriac, Geta Rîșnoveanu.

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
