## [Decision Letter · Decision Letter 0]

9 Sep 2019

PONE-D-19-21628

Wolf (Canis lupus, L. 1758) diet and prey selection in the South-Eastern Carpathian Mountains, Romania

PLOS ONE

Dear Mrs. Sin,

Thank you for submitting your manuscript to PLOS ONE. After careful consideration, we feel that it has merit but does not fully meet PLOS ONE’s publication criteria as it currently stands. Therefore, we invite you to submit a revised version of the manuscript that addresses the points raised during the review process.

Both reviewers found value in this study and provide numerous constructive comments to improve the MS.  I agree with those comments. Both reviewers asked to improve your English writing that would facilitate reading and improve the clarity of your message for readers. In addition, both make several specific suggestions in this regard, including new literature to be added to your MS.  I also agree with Rev #1 in in improving the methods section and avoiding speculative assessments of prey in diet of wolves.  The same with Rev #2 assessment, please follow reviewer recommendations.

We would appreciate receiving your revised manuscript by Oct 24 2019 11:59PM. To enhance the reproducibility of your results, we recommend that if applicable you deposit your laboratory protocols in protocols.io, where a protocol can be assigned its own identifier (DOI) such that it can be cited independently in the future. For instructions see: http://journals.plos.org/plosone/s/submission-guidelines#loc-laboratory-protocols

We look forward to receiving your revised manuscript.

Kind regards,

Paulo Corti, Ph.D.

Academic Editor

PLOS ONE

Journal Requirements:

2. In your Methods section, please provide additional location information of the sampling areas, including geographic coordinates for the data set if available.

4.  We suggest you thoroughly copyedit your manuscript for language usage, spelling, and grammar. If you do not know anyone who can help you do this, you may wish to consider employing a professional scientific editing service.  

Additional Editor Comments (if provided):

Reviewers' comments:

Reviewer's Responses to Questions

**Comments to the Author**

1. Is the manuscript technically sound, and do the data support the conclusions?

Reviewer #1: Yes

Reviewer #2: Yes

2. Has the statistical analysis been performed appropriately and rigorously? 

Reviewer #1: Yes

Reviewer #2: Yes

3. Have the authors made all data underlying the findings in their manuscript fully available?

Reviewer #1: Yes

Reviewer #2: Yes

4. Is the manuscript presented in an intelligible fashion and written in standard English?

Reviewer #1: No

Reviewer #2: Yes

5. Review Comments to the Author

Reviewer #1: This MS deals with the food habits and prey selection of the wolf in an area of the Carpathian mountains. It is an honest MS, with results which are new for that part of the wolf range, but not outstandingly new in absolute. I suggest to delete the scientific name of the wolf from the title. The methods are fine, although several clarifications are necessary (see comments below). Figures 2a and 2b are quite poor: I suggest changing them into histograms. A couple of relevant references are missing (see below). There are oversights here and there. The English is fair, although a few passages may require some “polishing”: a revision by a mother-tongue is needed.

LINE 26: insert “have” after “fragmentation”;

LINE 27: change “patched” into “patchy”;

LINE 58: change “microrodents” into “small rodents”;

LINE 61: change “focus” into “have focused”;

LINE 68: change “are” into “have been”;

LINE 75: change “characteristic to” into “representative of” or “common to”;

LINE 95: insert “has” before “led”;

LINE 101: just give the number of wolf scats analysed and drop those which were not: change “294” into “263”. Furthermore, it may be useful to indicate the month of each collecting visit, with the number of scats collected each time. As scats were collected at irregular intervals, sometimes only walking the itinerary twice/season, this information (which might be put in the online material) may help understand whether a season may be less well represented than the other;

LINES 123-124: discriminating scats of wolves from those of foxes only using their size (and possibly their shape) is tricky. Please, provide further details. Could the authors “test” their ability to discriminate by using DNA analyses e.g. on a sub-sample?

LINES 149-152: delete all the passage and move “(115 in winter, 121 in summer)” to line 191.

LINES 171 and 175-176: I have strong reservations on the methods which make use of formulas to estimate the ingested biomass of prey. In fact, it is usually impossible to know (i) whether a young/subadult/adult male/female has been preyed upon (body mass is normally quite different in different age classes and/or sexes, especially in polygynous mammals e.g. red deer and wild boar); (ii) whether other carnivores participated in the usage of the carcass; (iii) whether the carnivore fed alone on it or with conspecifics, e.g. a pair or even a pack. To limit uncertainties affecting the methods to calculate the biomass consumed from scats (cf. Chakrabarti et al. 2016; Lumetsberger et al. 2017), the estimated volume (Kruuk and Parish 1981) may still be the most reliable one, or at least the least unreliable, although it might be difficult to use it if scats tend to be made of just one food category. If formulas are used, their usage must be justified and the snags of this method should be pointed out clearly to caution the reader.

LINE 254: insert “t” in the term “randomization”;

LINE 260: please, use the term “small mammals” instead of “microrodents” and write in Italics the name “Martes”;

LINE 299: insert “it” after “modified”;

LINE 308: quote Mori et al. (2017) at the end of the sentence;

LINES 311-313: How were these densities calculated? Where do they come from? Which counting method was used? These is important information which should be indicated, even concisely – not only by mentioning a reference (besides, just a LIFE report).

LINE 320: insert Mori et al. (2017) at the end of the sentence.

LINE 350: change “increase” into “increased”,

LINE 386: I would eliminate “highly”: “inaccessible” is quite enough.

LINE 407: change “require” into “request”,

LINES 408-409: I would suggest to quote Imbert et al. (2016)’s findings here;

LINES 419-421: The conclusion of this MS is weak. I would delete it. Also delete the sub-heading “The dog and the domestic ungulates” (LINE 368) and make all part of the sub-heading “Seasonal diet”. Upon that, move LINES 380-390 to conclude the paper.

Chakrabarti S, Jhala YV, Dutta S, Qureshi Q, Kadivar RF, Rana VJ (2016) Adding constraints to predation through allometric relation of scats to consumption. J Anim Ecol 85:660–670.

Lumetsberger T, Ghoddousi A, Appel A, Khorozyan I, Walter M, Kiffner C (2017) Re-evaluating models for estimating prey consumption by leopards. J Zool 203:201-210.

I HAVE ATTACHED TO THIS REVIEW THE PDFs OF TWO MORE PAPERS.

Reviewer #2: Dear authors,

I have now read in detail and with pleasure your article entitled as “Wolf (Canis lupus, L. 1758) diet and prey selection in the South-Eastern Carpathian Mountains, Romania”

This is a very useful paper that adds value to the literature on wolf diet selection. I consider it of great interest for the international audience of the Journal, given also that it is the first systematic wolf diet study from Romania.

I have however some concerns regarding the structure and the linguistic quality of the text. Please improve your English and ask the help of a native English speaker to ameliorate your text. I have provided few corrections, but a thorough revision is needed.

The introduction needs substantial improvement in writing. It is very important to add your research questions as distinct research objectives at the end of the introduction. The methods and the results, as well the discussion should follow the order of the objectives.

Discussion needs substantial improvement and a reduction in word count by 30-40%. It needs to be shorter, clearer and more concise. Tables cannot appear in the discussion.

The reference list is not always updated and recent important citations are missing. It also needs shortening, by including only the papers that are up to the point. Avoid also over citation.

Some structural reform is needed as well. Please consider below my comments in detail, provided to improve the quality of the MS.

Line 6: Replace number of scats collected with the number of scats analyzed, as this is the number of scats that the analysis was based on

Line 8: Add a few lines in the abstract with the results for the most important prey species

Lines 20-22: Describe in a few sentences the general pattern of wolf diet with the appropriate references.

Line 22: There are recent papers from southern Europe indicating that wolves feed on garbage and livestock, e.g. Llaneza and Lopez-Bao 2015, Torres et al. 2015, Petridou et al. 2019

Lines 35-37: Difficult to understand, please rephrase

Lines 43-59: Move this part to methods. See also 90-91

Lines 61: I find this assumption a little too arbitrary. On the contrary, Newsome et al. 2016 said the opposite: “there is an urgent need to increase our understanding of grey wolf foraging ecology in human-dominated landscapes”.

Line 62: Once again, this assumption is a little arbitrary: “only a reduced number of studies address the large wolf populations living in complex natural ecosystems”, especially without a reference. Again, Newsome et al. 2016 state that “Most studies assessing the ecological role of grey wolves have been conducted in National Parks or wilderness areas, where grey wolves feed primarily on large wild ungulates”.

Lines 64-65: Please mention in a few words the main conclusion of the Romanian studies. You mention it in the discussion, I would like to see it in introduction as well.

Lines 66-69: this section should be expanded. Add some more info about your study: for example in what kind of area it was performed? For example: natural, anthropogenic area etch. The aims could be expanded and made more specific. Also, you could built up your hypotheses. For example, can you make predictions if your wolves will feed mostly on wild or domestic prey based on their populations in your study area? Also, any prediction on the species they would prefer? This would strengthen the paper.

Lines 74-76. Ecosystems and habitats …network. Text too general and not informative. Please delete, as you describe habitats later. Is your study area, or part of it, in a national park or similar? Please mention. If it is not all, please mention the percentage of it that is protected. Please mention the Natura 2000 site codes if any

Line 77: What you mean by fragmented? Fragmentation is a negative term, what described is more landscape heterogeneity.

Could you mention in your methods if hunting is allowed in your study area?

Lines 90-91: Please move the part 50-59 of Introduction in this methodological chapter of study area description. Please, include here data on species abundances or density, and use appropriate citations, and make text shorter. See also 95-98

Line 96: Rephrase herding units (e.g. livestock farms)

Lines 96-98. Please add a short description of the herds. E.g. are they permanent or transhumant? Are they accompanied by a shepherd and/or guardian dogs? Do they overnight outside or fenced? Are they close to villages or at higher pastures?

Please add data on the total number/density of livestock in the area. This is very important in order to compare with numbers of wild ungulates. Add the average herd size with SD. Please add a small description of the dogs in your area here, as it was an important diet item in your study. You have scattered info in the text. As a general but important comment, I would suggest to include this info in a table will all potential wolf prey species (wildlife and livestock/ dogs) with their relative abundances and relative references, when data available. All info should be organized and not scattered throughout the MS (please delete duplication).

Lines 102-103: was there an estimate of the wolf population after the study? Please add it and delete this sentence from here – it is not sampling methodology, move it to study area

Line 108: better replace “forestry roads” with “forest roads”

Line 128-131: I like the method of not collecting scats in 2km radius around active sheepfolds, although ad hoc selection. In line 130, please add “sheepfold areas inside the buffer zone”

Line 133: Were the samples stored in a freezer? In what temperature?

Line 140: Consider replacing “nails” with “claws”

Line 145: Consider replacing “operator” with “observer” or something similar

Line 147-148: Please provide a better description of blind test and the scores of the observers here.

Line 167: Consider adding Klare et al. 2011 as a reference for discarding the food remains with <5% of volume (it’s an important paper for diet studies)

Line 168. Please mention in the text that it is the correction factor of Weaver

Line 198-199: What is the difference between chi square test and (N-1) chi square test? Please clarify.

Lines 215-222: Please consider comment in Line 96-98. Just cite the table here. Biomass should not be a separate paragraph.

Line 226: comprised “of”

Line 233: Please comment here in your results the small participation of livestock in the diet of wolves.

Lines 233: Replace “micro-“ with “small”

Line 239: Add n=236 in the title of the table

Line 239-Table 2: Please sort results in the table by decreasing order of % occ. Besides, replace “indet*” with “undetermined” - confusing as it stands.

Line 250-252: Confusing text. Please rephrase. Include Fig. 1 where adequate.

Lines 286-306: Table 3 should be deleted from discussion. Introduce data in the Table 1, using appropriate references, provided that the same methodology is used in 1970 and 2012. If not such data exist, the table can’t be presented.

Line 299-306: This paragraph is difficult to follow.

Line 329-337: Snow cover doesn’t affect the other two species of ungulates as well?

Line 376: Rephrase “mongrel”, it’s confusing. For example: “small sized mixed-breed dogs”

Line 380-390: Please discuss also the availability of livestock compared to wild ungulates. When you provide their numbers/densities it will be more clear.

Line 398-399: 2016 hunting ban? What do you mean?

Line 400: are the animal losses per year?

Figures 2-3: Would the graphs be readable in the paper? It’s an interesting way to present the seasonal differences but I am afraid that they would be difficult to read in the final version.

6. PLOS authors have the option to publish the peer review history of their article (what does this mean?). If published, this will include your full peer review and any attached files.

Reviewer #1: Yes: Sandro LOVARI

Reviewer #2: Yes: Maria Petridou, University of Ioannina, Department of Biological Applications and Technology, Ioannina GR-45110, Greece

---

## [Author Response · Author response to Decision Letter 0]

25 Oct 2019

Dear Editor,

We are grateful for the positive comments of the referee to our manuscript and the helpful suggestions for correction which have helped us improve the paper. We have taken on board all of them.

We hope that the revised paper is now acceptable for publication but should there be any further issues please do not hesitate to contact us.

Sincerely,

Teodora Sin

Dear Mrs. Sin,

Thank you for submitting your manuscript to PLOS ONE. After careful consideration, we feel that it has merit but does not fully meet PLOS ONE’s publication criteria as it currently stands. Therefore, we invite you to submit a revised version of the manuscript that addresses the points raised during the review process.

Both reviewers found value in this study and provide numerous constructive comments to improve the MS. I agree with those comments. Both reviewers asked to improve your English writing that would facilitate reading and improve the clarity of your message for readers. In addition, both make several specific suggestions in this regard, including new literature to be added to your MS. I also agree with Rev #1 in in improving the methods section and avoiding speculative assessments of prey in diet of wolves. The same with Rev #2 assessment, please follow reviewer recommendations.

We would appreciate receiving your revised manuscript by Oct 24 2019 11:59PM. Please include the following items when submitting your revised manuscript:

• A rebuttal letter that responds to each point raised by the academic editor and reviewer(s). This letter should be uploaded as separate file and labeled 'Response to Reviewers'.

• A marked-up copy of your manuscript that highlights changes made to the original version. This file should be uploaded as separate file and labeled 'Revised Manuscript with Track Changes'.

• An unmarked version of your revised paper without tracked changes. This file should be uploaded as separate file and labeled 'Manuscript'.

2. In your Methods section, please provide additional location information of the sampling areas, including geographic coordinates for the data set if available.

Authors response: Geographic coordinates of the study sites were provided in the Methods section of the paper (Line 101 and Fig 1). Geographic coordinates for the entire data set will be available in the supporting information for the paper. 

Authors response: Information required was added in the Material and methods section of the MS (Lines 95-98), as follows: “Our research did not involve capture, handling or killing of animals, therefore did not require the approval of animal care and use procedures. Permissions for field studies were given by the Putna-Vrancea Natural Park Administration and Local Environmental Protection Agency.”

4. We suggest you thoroughly copyedit your manuscript for language usage, spelling, and grammar. If you do not know anyone who can help you do this, you may wish to consider employing a professional scientific editing service. 

Authors response: The MS was edited for language use and reviewed by a native English speaker (Hannah Kirkland) with relevant experience in the field. 

Authors response: Done

Authors response: Captions for Supporting information have been included in the revised MS and the materials will be provided as previously stated

Additional Editor Comments (if provided):

Reviewers' comments:

Reviewer's Responses to Questions

Comments to the Author

1. Is the manuscript technically sound, and do the data support the conclusions?

Reviewer #1: Yes

Reviewer #2: Yes

2. Has the statistical analysis been performed appropriately and rigorously?

Reviewer #1: Yes

Reviewer #2: Yes

3. Have the authors made all data underlying the findings in their manuscript fully available?

Reviewer #1: Yes

Reviewer #2: Yes

4. Is the manuscript presented in an intelligible fashion and written in standard English?

Reviewer #1: No

Reviewer #2: Yes

5. Review Comments to the Author

Reviewer #1: This MS deals with the food habits and prey selection of the wolf in an area of the Carpathian mountains. It is an honest MS, with results which are new for that part of the wolf range, but not outstandingly new in absolute.

Authors response: We are grateful for the positive comments on the paper and the extremely useful suggestions for further improvement.

I suggest to delete the scientific name of the wolf from the title. 

Authors response: Scientific name deleted.

The methods are fine, although several clarifications are necessary (see comments below). Figures 2a and 2b are quite poor: I suggest changing them into histograms. 

Authors response: changes made

A couple of relevant references are missing (see below). There are oversights here and there. 

Authors response: corrections made as suggested. 

The English is fair, although a few passages may require some “polishing”: a revision by a mother-tongue is needed.

Authors response: thank you! A native English speaker read the manuscript and it is now improved.

LINE 26: insert “have” after “fragmentation”; 

LINE 27: change “patched” into “patchy”;

LINE 58: change “microrodents” into “small rodents”;

LINE 61: change “focus” into “have focused”;

LINE 68: change “are” into “have been”;

LINE 75: change “characteristic to” into “representative of” or “common to”;

LINE 95: insert “has” before “led”;

Authors response: All these suggestions were accepted in the revised MS.

LINE 101: just give the number of wolf scats analysed and drop those which were not: change “294” into “263”. 

Authors response: 294 was changed to 236 (not 263), which is the correct number of valid samples

Furthermore, it may be useful to indicate the month of each collecting visit, with the number of scats collected each time. As scats were collected at irregular intervals, sometimes only walking the itinerary twice/season, this information (which might be put in the online material) may help understand whether a season may be less well represented than the other;.

Authors response: All this information will be provided as Supporting information in S1 Table.

LINES 123-124: discriminating scats of wolves from those of foxes only using their size (and possibly their shape) is tricky. Please, provide further details. Could the authors “test” their ability to discriminate by using DNA analyses e.g. on a sub-sample?

Authors response: The samples collected for this study were not genetically analysed. Nevertheless, in a subsequent study undertaken during 2014-2017 (LIFE13NAT/RO/000205), the DNA analyses of 187 scat samples collected and classified following the same criteria of differentiation we used in the present study, by the same operators as in our study, confirmed their ability to discriminate between wolf and fox scats. In 97% of the cases genetic analysis confirmed that scats collected belong to wolves. The rest of 3% of the samples belonged to other large carnivores and none of the samples belonged to fox. 

The text was modified to (lines 166-169): “The DNA analyses of 187 scat samples collected by the same observers in a subsequent study confirmed their ability to discriminate between wolf and other carnivores scats, with 97% (n = 181) of the samples being correctly attributed to wolf.”

LINES 149-152: delete all the passage and move “(115 in winter, 121 in summer)” to line 191.

Authors response: paragraph deleted and “(115 in winter, 121 in summer)” moved to line 133.

LINES 171 and 175-176: I have strong reservations on the methods which make use of formulas to estimate the ingested biomass of prey. In fact, it is usually impossible to know (i) whether a young/subadult/adult male/female has been preyed upon (body mass is normally quite different in different age classes and/or sexes, especially in polygynous mammals e.g. red deer and wild boar); (ii) whether other carnivores participated in the usage of the carcass; (iii) whether the carnivore fed alone on it or with conspecifics, e.g. a pair or even a pack. To limit uncertainties affecting the methods to calculate the biomass consumed from scats (cf. Chakrabarti et al. 2016; Lumetsberger et al. 2017), the estimated volume (Kruuk and Parish 1981) may still be the most reliable one, or at least the least unreliable, although it might be difficult to use it if scats tend to be made of just one food category. If formulas are used, their usage must be justified and the snags of this method should be pointed out clearly to caution the reader.

Authors response: As in over 95% of the total samples we identified only one food item, we decide to keep the focus on the biomass and made the clarification as the reviewer suggested. We added the following text (lines 201-203): “Considering that the degree of digestion and digestibility is different for every food item [25], and that Vm overestimates the importance of prey when the percentage of scats containing only one item is high, … .” and (lines 209-215): “Although assessing carnivore diet using biomass deals with the drawbacks of ranking food items by frequency and volume measures [25], estimating ingested biomass of prey from scats is still prone to bias because the relationship between the fresh mass of prey per scat and prey body mass is influenced by some variables that in nature are hard to detect and measure. Among these we highlight: the age class or sex of prey, the amount and carcass body parts consumed by wolf, and the number of conspecifics feeding on the same carcass [36, 37].”

LINE 254: insert “t” in the term “randomization”;

LINE 260: please, use the term “small mammals” instead of “microrodents” and write in Italics the name “Martes”;

Authors response: corrections made. “Microrodents” changed to “small rodents”, as suggested 

LINE 299: insert “it” after “modified”;

Authors response: sentence rephrased (lines: 331-332)

LINE 308: quote Mori et al. (2017) at the end of the sentence;

Authors response: Mori et al. (2017) was added

LINES 311-313: How were these densities calculated? Where do they come from? Which counting method was used? These is important information which should be indicated, even concisely – not only by mentioning a reference (besides, just a LIFE report).

Authors response: supplementary information was provided in the Materials and methods section in the Data analysis (lines 258-268)

LINE 320: insert Mori et al. (2017) at the end of the sentence.

LINE 350: change “increase” into “increased”,

LINE 386: I would eliminate “highly”: “inaccessible” is quite enough.

LINE 407: change “require” into “request”,

LINES 408-409: I would suggest to quote Imbert et al. (2016)’s findings here;

Authors response: corrections made

LINES 419-421: The conclusion of this MS is weak. I would delete it. Also delete the sub-heading “The dog and the domestic ungulates” (LINE 368) and make all part of the sub-heading “Seasonal diet”. Upon that, move LINES 380-390 to conclude the paper.

Authors response: Thank you! Corrections made as requested

Chakrabarti S, Jhala YV, Dutta S, Qureshi Q, Kadivar RF, Rana VJ (2016) Adding constraints to predation through allometric relation of scats to consumption. J Anim Ecol 85:660–670.

Lumetsberger T, Ghoddousi A, Appel A, Khorozyan I, Walter M, Kiffner C (2017) Re-evaluating models for estimating prey consumption by leopards. J Zool 203:201-210.

I HAVE ATTACHED TO THIS REVIEW THE PDFs OF TWO MORE PAPERS.

Authors response: Thank you! References were added.

Reviewer #2: Dear authors,

I have now read in detail and with pleasure your article entitled as “Wolf (Canis lupus, L. 1758) diet and prey selection in the South-Eastern Carpathian Mountains, Romania”

This is a very useful paper that adds value to the literature on wolf diet selection. I consider it of great interest for the international audience of the Journal, given also that it is the first systematic wolf diet study from Romania.

Authors response: We are grateful for the positive comments on the paper and the extremely useful suggestions for further improvement.

I have however some concerns regarding the structure and the linguistic quality of the text. Please improve your English and ask the help of a native English speaker to ameliorate your text. I have provided few corrections, but a thorough revision is needed.

Authors response: thank you. A native English speaker read the manuscript and it is now improved.

The introduction needs substantial improvement in writing. It is very important to add your research questions as distinct research objectives at the end of the introduction. The methods and the results, as well the discussion should follow the order of the objectives.

Authors response: the research objectives were rewritten and the Methods, Results and Discussions sections were structured accordingly.

Discussion needs substantial improvement and a reduction in word count by 30-40%. It needs to be shorter, clearer and more concise. 

Authors response: corrections made as suggested. The Discussion is now shorter, even if additional information was added. 

Tables cannot appear in the discussion.

Authors response: Table deleted from the discussion section and will be provided as Supporting information (S2 Table).

The reference list is not always updated and recent important citations are missing. It also needs shortening, by including only the papers that are up to the point. Avoid also over citation.

Authors response: Thank you! Correction made as requested.

Some structural reform is needed as well. Please consider below my comments in detail, provided to improve the quality of the MS.

Line 6: Replace number of scats collected with the number of scats analyzed, as this is the number of scats that the analysis was based on

Authors response: corrected

Line 8: Add a few lines in the abstract with the results for the most important prey species

Authors response: Addition was made as requested. Please see lines 30-35.

Lines 20-22: Describe in a few sentences the general pattern of wolf diet with the appropriate references.

Authors response: The following text was added (lines 42-46): “[1]. In Europe, recent reviews revealed that wolf feeds mainly on medium-sized wild ungulates, such as wild boar Sus scrofa, roe deer Capreolus capreolus and chamois Rupicapra rupicapra or large-sized wild ungulates, such as reindeer Rangifer tarandus, elk Alces alces and red deer Cervus elaphus [2, 3].”

Line 22: There are recent papers from southern Europe indicating that wolves feed on garbage and livestock, e.g. Llaneza and Lopez-Bao 2015, Torres et al. 2015, Petridou et al. 2019

Authors response: Thank you! References have been added as suggested

Lines 35-37: Difficult to understand, please rephrase

Authors response: text was rewritten (please see lines 61-63)

Lines 43-59: Move this part to methods. See also 90-91

Authors response: The lines from 50 to 59 were moved to the Methods section where they replaced the lines 90-91 (now the information may be found in Lines (116-121). We chose to keep the information in lines 48-55 in the Introduction section as it gives an overview of the wolf spatial and temporal distribution at the national scale and not in the study region. Highlighting the reduction of the wolf habitat across the country enforces the relevance of our research.

Lines 61: I find this assumption a little too arbitrary. On the contrary, Newsome et al. 2016 said the opposite: “there is an urgent need to increase our understanding of grey wolf foraging ecology in human-dominated landscapes”.

Line 62: Once again, this assumption is a little arbitrary: “only a reduced number of studies address the large wolf populations living in complex natural ecosystems”, especially without a reference. Again, Newsome et al. 2016 state that “Most studies assessing the ecological role of grey wolves have been conducted in National Parks or wilderness areas, where grey wolves feed primarily on large wild ungulates”.

Authors response: text was corrected and now it refers only to research available for the Romanian part of the wolf population (please see Lines 77-79).

Lines 64-65: Please mention in a few words the main conclusion of the Romanian studies. You mention it in the discussion, I would like to see it in introduction as well.

Authors response: we added (Lines 80-82): “Based on stomach content analyses, in the 1970s wild and domestic ungulates occurred in similar proportions (28% and 27%, respectively) [19], while more recently higher percentages of wild ungulates (53%) were reported [20].”

Lines 66-69: this section should be expanded. Add some more info about your study: for example in what kind of area it was performed? For example: natural, anthropogenic area etch. The aims could be expanded and made more specific. Also, you could built up your hypotheses. For example, can you make predictions if your wolves will feed mostly on wild or domestic prey based on their populations in your study area? Also, any prediction on the species they would prefer? This would strengthen the paper.

Authors response: Thank you for suggestion. Text was edited accordingly (lines: 83-93)

Lines 74-76. Ecosystems and habitats …network. Text too general and not informative. Please delete, as you describe habitats later. Is your study area, or part of it, in a national park or similar? Please mention. If it is not all, please mention the percentage of it that is protected. Please mention the Natura 2000 site codes if any

Authors response: The non-informative text was deleted, and the codes of the Natura 2000 sites were added (Line 81).

Line 77: What you mean by fragmented? Fragmentation is a negative term, what described is more landscape heterogeneity.

Authors response: corrected. The word was deleted (Line 102).

Could you mention in your methods if hunting is allowed in your study area?

Authors response: Yes. Hunting is allowed. The information was added in the Methods section (Lines 123-125).

Lines 90-91: Please move the part 50-59 of Introduction in this methodological chapter of study area description. 

Authors response: Changes done. See the comment above for the same request;

Please, include here data on species abundances or density, and use appropriate citations, and make text shorter. See also 95-98

Authors response: The text was shortened and data on species abundance were provided in Table 1. Details on wild and domestic ungulate data collection and estimation of their population size was provided in the Materials and methods section as requested by the reviewer #1 (Line 258-268). 

Line 96: Rephrase herding units (e.g. livestock farms)

Authors response: as the transhumance practice is still present, and in the region are not real farms, we consider that herd units reflect better the characteristics of the area.

Lines 96-98. Please add a short description of the herds. E.g. are they permanent or transhumant? Are they accompanied by a shepherd and/or guardian dogs? Do they overnight outside or fenced? Are they close to villages or at higher pastures?

Please add data on the total number/density of livestock in the area. This is very important in order to compare with numbers of wild ungulates. Add the average herd size with SD. Please add a small description of the dogs in your area here, as it was an important diet item in your study. 

Authors response: Addition was made as suggested (Line 126-131) and numbers of domestic species are presented in Table 1.

You have scattered info in the text. As a general but important comment, I would suggest to include this info in a table will all potential wolf prey species (wildlife and livestock/ dogs) with their relative abundances and relative references, when data available. All info should be organized and not scattered throughout the MS (please delete duplication).

Authors response: Thank you! The data were included in Table 1 and in the text we refer to it as appropriate.

Lines 102-103: was there an estimate of the wolf population after the study? Please add it and delete this sentence from here – it is not sampling methodology, move it to study area

Authors response: In fact this was a kind of information which determined our sampling strategy. Sentence was rephrase so that to make this connection clear (Lines 133-136). 

Line 108: better replace “forestry roads” with “forest roads”

Authors response: corrected

Line 128-131: I like the method of not collecting scats in 2km radius around active sheepfolds, although ad hoc selection. In line 130, please add “sheepfold areas inside the buffer zone”

Authors response: correction made

Line 133: Were the samples stored in a freezer? In what temperature?

Authors response: the samples were kept in a cold storage room, at a constant temperature of 4⁰C. The information was added in the MS (Line 170).

Line 140: Consider replacing “nails” with “claws”

Line 145: Consider replacing “operator” with “observer” or something similar

Authors response: corrected

Line 147-148: Please provide a better description of blind test and the scores of the observers here.

Authors response: the following text was added (lines 179-184) “Blind tests were applied on randomly selected samples from the available collection of hairs of wild and domestic mammal species present in our study area to assess the ability of the two observers to identify the prey species. The process was repeated at least three times, and a species was considered to be accurately determined if the responses of both observers matched in 11 out of 12 cases (92%). To further increase the reliability, all the collected samples were cross-checked by both observers.

Line 167: Consider adding Klare et al. 2011 as a reference for discarding the food remains with <5% of volume (it’s an important paper for diet studies)

Authors response: The suggested reference was added.

Line 168. Please mention in the text that it is the correction factor of Weaver

Authors response: correction made

Line 198-199: What is the difference between chi square test and (N-1) chi square test? Please clarify.

Authors response: According to Campbell (2007), (N-1) chi square test performs better when sample sizes are small due to rarity of a condition (i.e. rare prey items). This test and corresponding paragraphs were removed from the MS because we decided to focus our discussion on the diet expressed as ingested biomass.

Lines 215-222: Please consider comment in Line 96-98. Just cite the table here. Biomass should not be a separate paragraph.

Authors response: corrections made as suggested

Line 226: comprised “of”

Authors response: the native English speaker who check our MS prefer the initial version, without “of” 

Line 233: Please comment here in your results the small participation of livestock in the diet of wolves.

Authors response: information added as suggested (lines: 277-280).

Lines 233: Replace “micro-“ with “small”

Line 239: Add n=236 in the title of the table

Line 239-Table 2: Please sort results in the table by decreasing order of % occ. Besides, replace “indet*” with “undetermined” - confusing as it stands.

Authors response: corrections made

Line 250-252: Confusing text. Please rephrase. Include Fig. 1 where adequate.

Authors response: text was edited for better clarity (please see Lines 294-296)

Lines 286-306: Table 3 should be deleted from discussion. Introduce data in the Table 1, using appropriate references, provided that the same methodology is used in 1970 and 2012. If not such data exist, the table can’t be presented.

Authors response: Table 3 was deleted from the discussion section and moved as a Supporting information (S2 Table). All the data included in the table were calculated based on the official raw data reported by the Romanian authorities, in 1970 and 2012. 

Line 299-306: This paragraph is difficult to follow.

Authors response: editing was done and the clarity was improved (please see Lines 331-339)

Line 329-337: Snow cover doesn’t affect the other two species of ungulates as well?

Authors response: It does, but as mentioned in the text, not at the same extent: “The average snow depth of 50 cm in our region may affect wild boar escape speed, while the red deer is supposedly able to flee faster, given the high stature and length of the limbs”. 

Line 376: Rephrase “mongrel”, it’s confusing. For example: “small sized mixed-breed dogs”

Authors response: correction made

Line 380-390: Please discuss also the availability of livestock compared to wild ungulates. When you provide their numbers/densities it will be more clear.

Authors response: Information on availability of livestock referred to in the discussion. However, comparing the wild and domestic ungulates availability in summer is not possible because summer estimates of wild ungulates are not available (these aspects were clarified in the Materials and methods section).

Line 398-399: 2016 hunting ban? What do you mean?

Authors response: Before October 2016, the Romania’s Ministry of Environment annually issued harvest quotas for large carnivores. After that time the hunting was completely interdicted (banned) due to concerns about the uncertainties related to monitoring protocols and quality of the abundance estimates of large carnivores populations in Romania.

Line 400: are the animal losses per year?

Authors response: animal losses are for one grazing season. The information was added in the text (line 413).

Figures 2-3: Would the graphs be readable in the paper? It’s an interesting way to present the seasonal differences but I am afraid that they would be difficult to read in the final version.

Authors response: We followed the recommendation of Reviewer #1 and changed the presentation of Figures 2a and 2b.

6. PLOS authors have the option to publish the peer review history of their article (what does this mean?). If published, this will include your full peer review and any attached files.

Do you want your identity to be public for this peer review? For information about this choice, including consent withdrawal, please see our Privacy Policy.

Reviewer #1: Yes: Sandro LOVARI

Reviewer #2: Yes: Maria Petridou, University of Ioannina, Department of Biological Applications and Technology, Ioannina GR-45110, Greece

---

## [Editor Report · Decision Letter 1]

6 Nov 2019

Wolf diet and prey selection in the South-Eastern Carpathian Mountains, Romania

PONE-D-19-21628R1

Dear Dr. Sin,

We are pleased to inform you that your manuscript has been judged scientifically suitable for publication and will be formally accepted for publication once it complies with all outstanding technical requirements.

With kind regards,

Paulo Corti, Ph.D.

Academic Editor

PLOS ONE

---

## [Editor Report · Acceptance letter]

13 Nov 2019

PONE-D-19-21628R1 

Wolf diet and prey selection in the South-Eastern Carpathian Mountains, Romania 

Dear Dr. Sin:

I am pleased to inform you that your manuscript has been deemed suitable for publication in PLOS ONE. Congratulations! Your manuscript is now with our production department. 

With kind regards,

on behalf of

Dr. Paulo Corti 

Academic Editor

PLOS ONE